# Deep Clustering and Interpolation via the Federated Self-Organizing Map

## Abstract

We introduce FedSOM, a clustering and interpolation module based on the Self-organizing Map (SOM), which can be appended to any encoder and which can be trained in a federated way either in tandem with the encoder or post training on the resulting representations. The result is a discrete moduli space of representations that provides for cluster or sample-level interpolation, hierarchical clustering, and can be leveraged as a function to cluster new vectors at test time. This moduli space can either be created from data alone or by glueing pre-existing clusters along regions of commonality, although we do not explore the latter in this work. Interpolation is accomplished by considering the $n$-dimensional tensor underlying the SOM as a weighted undirected graph, where the weights are computed as a function of the dispersion of the two clusters corresponding to the nodes bounding the given edge. Any two clusters or samples may then be interpolated by computing the lowest-cost path between their associated graph nodes via Dijkstra's algorithm. The method is validated on MNIST-like and parsed-binary malware datasets.

## 1 Introduction

Self-supervised learning has been successfully applied to clustering and interpolation tasks in domains such as vision, text, and speech, in which datasets are large and homogeneous. However, recent advancements such as VIME [27] have facilitated the application of self-supervised learning to clustering and interpolation on tabular data. The typical approach to creating representations for downstream tasks is to minimize a contrastive loss, in which positive pairs are constructed by conditionally sampling from column marginals. Non-contrastive methods such as VICReg [2] have also shown great promise on downstream image classification tasks. Approaches to deep clustering vary widely not only in terms of preferred data, but in terms of classical cluster approaches, i.e., centroid based, density based, and online vs batch. In [44] they explore clustering utterance data with BERT embeddings with a centroid-based approach. Works such as [8] and [29] rely on a softmax function to assign cluster labels, meaning that the number of clusters must be fixed at train time. The number of clusters are also fixed at train time in [17]. While the vast majority of deep clustering methods are batch trained, online methods such as [45] also exist, in which the number of clusters need not be set at train time. In [48] the authors describe a GMM-type approach in which the number of clusters is cast as a hidden variable in an EM-type approach. Similar to multi-modal learning on tabular data, multi-view learning has successfully been applied to both tabular and feature-homogeneous datasets as in [46] and [47], in which multiple views take the place of tabular perturbations. The latter seeks to discover correlations between multiple views of the same object. This differs from our work in that perturbations are created for the sake of creating tightly-grouped representations for downstream clustering tasks.

Self-organizing map structures have long been leveraged both for clustering and interpolation. Training methodologies vary widely, but the main idea is that SOM tensor nodes correspond to cluster labels. In [9], the authors construct SOM-VAE in which they combine the structure of a SOM with a differentiable learning objective to learn the underlying discrete structure. Learning such a discrete structure is an example of vector quantisation. See [23] in which a vector quantised-variational autoencoder is constructed for homogeneous (images, video, speech) rather than tabular data. Interpolation is one of the primary applications of self-organizing map structures and there are many such approaches. Leveraging weighted-averages between weight vectors attached to SOM tensor nodes is

a popular approach taken in [10], in which they leverage topological information contained in local neighborhoods within the SOM to learn nonlinear relationships between samples in the input data. A additional layer may also be added as in [11] that computes interpolations between two given nodes. Interpolation may also be carried out by leveraging the entire SOM. This sort of approach is appropriate when a SOM is trained for each sample in an image dataset [13]. Retraining is not typically required for SOM-based interpolation. See [26] for further details. Interpolation is also leveraged to aid in the learning of deep networks, as in [21] in which interpolation is leveraged to learn intermediate representations of time series samples.

## 1.1 OUR CONTRIBUTION

- A novel self-organizing map structure that can be trained in a federated way
- A graph structure underlying our self-organizing map, which can be leveraged for cluster interpolation in a way that does not depend on the manifold assumption
- A moduli space of cybersecurity data that allows analysts to visualize relationships, perform interpolation-based parser analysis, and perform similarity queries that are conditioned on neighboring families

### 1.1.1 LIMITATIONS

The number of nodes in the SOM tensor $\tau$ is exponential in the dimension of the SOM, so training becomes prohibitive for $\dim \tau \geq 4$ when the number of samples exceeds 1e5. Because training is performed in an unsupervised way, the contrastive learning objective can lead to label-inhomogeneous clusters and over-partitioning of coherent clusters. SOM training is performed classically, meaning that the method does not benefit from being structured as a loss minimization problem. This also represents a benefit, however, in that the method can be applied to existing representations.

Finally, malware representations result from feeding parsed binaries through the encoder, which means information available to FedSOM is limited by the parser itself. Future work should include representations created from raw bytes.

## 2 RELATED WORK

In this work we define interpolation between two points as identifying a finite sequence of samples that are successively similar and represent a transformation from the first sample to the second. We do not refer to *interpolative* points as in [23,25] in which a point is said to be interpolative if it lies within the convex hull of the given dataset. Most prior art leveraging self organizing maps for interpolation involves the construction a sequence of self organizing maps that functions as a discrete set of intermediate steps between two given samples. This differs dramatically from our approach in which interpolation takes place within a fixed SOM. The purpose of leveraging a self-organizing map is to create a discrete topological space to represent a typically continuous latent space, or to learning relationships between classes, as in [5]. As in the case of [14], the SOM clustering during training encourages convex clusters. As in [14], we considered training the encoder in tandem with the clustering algorithm, but settled on training the SOM on trained representations for wider method applicability. There is no effort devoted to learning illusory underlying manifolds as in [28] and instead focus on learning proximity-based cluster assignments via the SOM. In [7], they construct smooth interpolations in latent space to construct interpolations via GANs in the space of architectural design. Our notion of interpolation is akin to the *Category Morphing* process described in [15]. Self-organizing maps have been used previously for interpolation via stepping successively between neighboring nodes. This method was leveraged in [9] for interpolating between steps in a time series. Interpolation between SOM nodes is performed in [9] by computing a weighted average between weight vectors, where the weight is a function of the given input vectors. This is essentially a generative method in that representation vectors which do not correspond to any samples in the original dataset are leveraged for interpolation. In contrast, all interpolations in this work are performed using only samples that exist in the data.

The application of these methods to cybersecurity data is non-existent. Our self-organizing map approach requires the user to specify only an upper bound on the number of clusters by specifying

the dimensions of the underlying tensor. Small perturbations in the encoder will change the inferred metric on representation space, and therefore alter the geodesic interpolative path connecting two test points. We avoid this sensitivity by creating a discrete moduli space of the input data optimized for clustering and interpolation. Current SOM-based interpolation approaches involve either the discretization of each sample by training a SOM and considering the sequence of such, or by computing a weighted average of weight vectors attached to neighboring nodes.

## 3  PRELIMINARIES

For the purpose of creating representations on which FedSOM is trained, we train both UMAP and neural network encoders, the latter by minimizing some form of self-supervised loss. We take the term *interpolation* to refer to a transformational sequence of samples that connects two given samples, rather than referring to the well-known mode of overfitting, as described in [3].

### 3.1  SELF-ORGANIZING MAPS

A self-organizing map is a collection of learnable weight vectors indexed by the coordinates of a tensor, along with an update rule that encourages the weight vectors to closely match the input vectors. This update rule includes a neighborhood function that determines the extent to which weight vectors attached to neighbor nodes are updated. See 1 for a visual representation. The key SOM hyperparameter is $\sigma$, the variance of the Gaussian kernel which determines the reach of the neighborhood function. The result is a map of the input data such that weight vector similarity recedes with distance within the indexing tensor.

### 3.2  DATA

Interpolation and clustering of cybersecurity data is the focus of this work, but the method itself can be applied to a variety of sparse tabular datasets. In order to validate this claim, we train our models not only on parsed binaries and network data, but on MNIST-like datasets. This latter family was desirable for its sparsity and its well-defined structural and ontological groupings.

#### 3.2.1  CYBERSECURITY

Parsed portable executable datasets based on a parser are not amenable to smooth interpolation within representation space. It is highly unlikely that a given region in latent space could give rise to feature vectors that could have been produced by parsing executable binaries. For this reason, interpolation is performed by considering only representations corresponding to samples in the corpus.

In the case of Ember and Sorel20M, we considered only feature vectors corresponding to malicious binaries. This was done to focus our experiments, but future work should include clustering and interpolation with clean files included. In this way one may be able to discover and understand parasitic file infectors from the point of view of the leveraged parser.

## 4  METHOD

FedSOM is a federated self-organizing map with an underlying weighted graph structure that facilitates clustering and interpolation between any two clusters or any two samples. Interpolation between clusters of parsed binaries can be leveraged to understand the parser by inspecting feature distribution changes through the interpolation path.

We consider two approaches to embedding creation for the sake of FedSOM clustering and interpolation. The first is an encoder with a self-supervised learning objective. This approach benefits from the scalability and flexibility of batch training. The second is UMAP, which offers the advantage of considering all data at once and allows for a globally optimized topological embedding for datasets that satisfy the manifold hypothesis. See [1] for code and data.

---

[1]https://github.com/fed-som/fedsom, https://zenodo.org/records/11205063

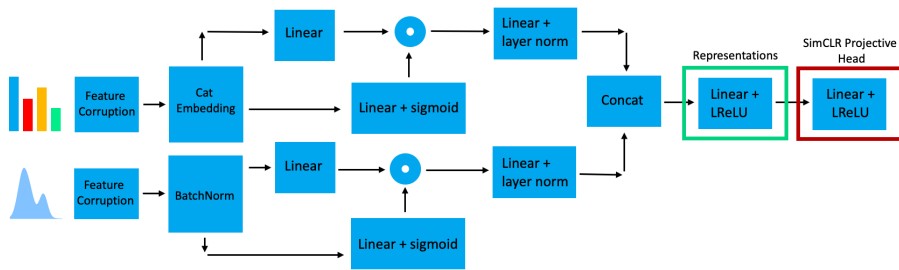

Figure 1: Categorical columns are first separated from continuous columns either via a heuristic. The categorical columns are separately passed through embedding layers. The remainder of the network is residual with a projection head for use in the contrastive learning objective.

## 4.1 NEURAL NETWORK ENCODER

The encoder is a vanilla residual network that separately handles continuous and categorical values. The categorical variables are discovered by an initial sweep through the data, based on a set of heuristics. We cast a column as categorical if any one of the following holds: (1) contains only bools, (2) contains only integers, (3) contains at least one string.

Categorical columns are encoded by an integer label encoder, and then embedded into a low-dimensional Euclidean space via an embedding layer $\mathcal{E}$. These embeddings are concatenated and passed through two separate fully-connected layers. The output of the first passes through a sigmoid and is combined with the latter via a Hadamard product. The result then passes through a third fully connected layer after which we apply layer norm.

Replacing the encoder and embedding layer with batchnorm, the numerical columns pass through an identical network. The numerical and categorical embeddings are then concatenated and passed through two fully-connected + leaky relu layers, the latter function as the SimCLR projection head.

All loss functions, with the exception of VICReg, are contrastive in nature. This means that for a given batch $x$, the loss is a function of the form $\mathcal{L}(x, \tilde{x})$, where $\tilde{x}$ is a perturbed version of $x$ and $(x, \tilde{x})$ is termed a *positive* pair. For the sake of constructing positive pairs, we leverage a perturbations module.

### 4.1.1 TABULAR PERTURBATIONS MODULE

Perturbation is performed by conditionally drawing from the marginal distribution of each column, which is a uniform over $[\min, \max]$ if the column is numerical and a uniform over unique observations if the column is categorical. The uniform distribution was chosen for the sake of maximizing sample speed.

We first perform a distribution learning step in which the sparsity of each column is measured via Welford's algorithm applied counts of $x.\text{is\_null}() \,|\, (x == 0)$. Columns with sparsity above a set threshold are discarded before being passed to the encoder. An example of such a column would be the upper left pixel of the MNIST dataset.

We perturb batches according to the following recipe:

- remove columns above the *upper* sparsity threshold $\psi_u$
- for each column $c$, draw a Bernoulli random variable $\rho_c \sim B(1, p)$ for $p$ fixed across columns
- if $\rho_c = 1$, draw $\text{batch\_size}$ samples from the marginal distribution $P(c)$ of column $c$
- if the sparsity of $c$ is above the *lower* sparsity threshold $\psi_b$, then draw $\text{batch\_size}$ random variables $\rho_r \sim B(1, q)$ and for each $\rho_r = 1$, sample from $U(\Omega_c)$ if $c$ is categorical and from $U([\min(\Omega_c), \max(\Omega_c)])$ if c is numerical.

The purpose of the final step is to ensure contrastive positive pairs differ on sparse columns. The probability of perturbing a given element with table coordinates $(r, c)$ is then $pq$ if $\text{sparsity}(c) > \psi_b$ and $p$ otherwise. The values $\psi_u, \psi_b, p, q$ are chosen via hyperparameter tuning.

### 4.1.2 LOSS FUNCTIONS

Depending on the dataset, we utilize one of the following loss functions: SimCLR [4], VICReg [32], C3 [18], or our own loss function ContVICReg to create representations for the sake of applying FedSOM.

For each batch $x$, ContVICReg is defined as $\text{SimCLR}(x) + \sum_{C_j \in \mathcal{C}_z} \mathcal{L}_{\text{vic}}(C_j)$, where $\mathcal{C}_z$ is the set of clusters obtained by training HDBSCAN on the pre-projection head representations $z$.

## 4.2 UMAP ENCODER

For the sake of testing FedSOM on embeddings created independently of any sort of SOM structure, we leverage UMAP on smaller datasets. We train separate UMAP encoders for categorical and numerical columns. A third UMAP encoder is then trained on the concatenation of the output from the first two. HDBSCAN is leveraged for optimization by maximizing normalized mutual information between the given cluster labels and the labels learned by HDBSCAN.

Datasets are preprocessed by $Z$-score normalization applied to the nonzero rows of each numerical column to preserve the sparsity of the original dataset.

## 4.3 SELF ORGANIZING MAP

The Self-Organizing Map update rule is given by $w_i^{s+1} = w_i^s + \theta(\beta, i)\alpha(v - w_i^s)$, where $v$ is the sample, $\theta$ is the neighborhood kernel function, $\beta$ is the index of the best matching unit (node) in the som grid, $i$ is the index of the node being updated, $s$ is the current iteration, $\alpha$ is the learning rate, $w_i^s, w_i^{s+1}$ are and the old and new weight vectors for node $i$, respectively. The neighborhood function is given by $\theta(\beta, i) = \exp(-\|\tau[i] - \tau[\beta]\|_2 / 2\sigma^2)$, where $\tau$ is the tensor of node coordinates. See Algorithm 2 for details.

The update rule for FedSOM is similar, but differs in that the training corpus consists of the weights of the several SOMs.

### 4.3.1 TENSOR DIMENSIONALITY

The ability of the SOM to partition a neighborhood of representation space into distinct clusters is a function of the number of nodes neighboring a given node in the tensor. The maximum number of distinct clusters neighboring a given cluster is equal to $3^{\dim(\tau)} - 1$, making train time exponential in the number of tensor dimensions. We thus limited our experiments to dimensions two and three.

## 4.4 THE FEDERATED SOM

The Federated Self Organizing Map is a natural extension of the SOM, as it consists of nothing more than a set of disparate SOMs along with an additional *meta* SOM trained on the weight vectors of the individual SOMs. The original dataset $D$ is partitioned by i.i.d. sampling to ensure that each SOM is trained on a subset drawn from the same distribution as the original dataset.

Training is performed in a federated way in the sense that separate SOM models are learned for each partition of the original data.

---

**Algorithm 1:** Federated Self-Organizing Map

---

**Result:** $\tau_j, W_j = \mathrm{som}_j(D_j), \tau_m, W_m = \mathrm{som}_m(\bigsqcup W_j)$ where $j$ indexes the i.i.d. sampled partition of the dataset $D$, and $\mathrm{som}_m$ is the *meta* SOM operating on the weight vector sets $W_j$ of the several SOMs.

**Data:**

- $D$: the dataset
- $N_s$: number of SOMs
- Remaining parameters as in Algorithm 1, uniform across $\{\mathrm{som}_j\}$

Partition $D = \bigsqcup D_j$ into $N_s$ subsets by i.i.d. sampling;

**for** $j$ *in* $[1, \ldots, N_s]$ **do**
  |     train $\tau_j, W_j = \mathrm{som}_j(D_j)$
**end**
train $\tau_m, W_m = \mathrm{som}_m(\bigsqcup W_j)$

---

### 4.4.1 THE FEDSOM GRAPH

We construct a graph from the SOM tensor in order to create a structure within which any two clusters or any two samples may be interpolated. Weights are defined so as to facilitate interpolation between neighboring nodes corresponding to high-quality clusters, i.e., clusters with low dispersion and high pairwise separation.

The nodes of the graph correspond to the nodes of the tensor and two nodes are connected by an edge in the graph if and only if the nodes are neighbors in the tensor. Two nodes with coordinates $(n_1, \ldots, n_{\dim(\tau)}), (m_1, \ldots, m_{\dim(\tau)})$ are neighbors if and only if $\sum_j |n_j - m_j| = 1$.

The Calinski-Harabazs score is given by

$$\kappa = \frac{(N - k) \sum_j^k n_j \left\| \mathbf{c}_j - \mathbf{c}_{\text{total}} \right\|^2}{(k - 1) \sum_j^k \sum_{x_i \in C_j} \left\| x_i - \mathbf{c}_j \right\|^2}, \tag{1}$$

where $N$ is the number of samples, $k$ is the number of clusters, $n_j$ is the number of data points in cluster $j$, $\mathbf{c}_j$ is the centroid of cluster $j$, $\mathbf{c}_{\text{total}}$ is the centroid of the entire dataset, and $C_j$ is the set of data points belonging to cluster $j$.

Weights are assigned to the edges based on the Calinski-Harabazs score computed from the embeddings attached to the two nodes bounding the given edge. The weight assigned to the edge $e_{uv} := e(u, v)$ is

$$\omega(e_{uv}) = \begin{cases} 1 - (\kappa / (\kappa + 1)) & \text{if } |\mathbf{c}_u|, |\mathbf{c}_v| > 2 \\ 0 & \text{otherwise} \end{cases} \tag{2}$$

where $\kappa$ is computed only from the samples whose cluster labels are given by the nodes $u, v$, and $\mathbf{c}_u, \mathbf{c}_v$ are the clusters corresponding to nodes $u, v$, respectively.

Edge weights are normalized so as to sit within $[0, 1)$, as the Calinski-Harabazx score is unbounded above. Because edge weight corresponds to cost in the utilized version of Dijkstra's algorithm, we subtract the normalized score from 1 as large $\kappa$ indicates high cluster separation and low intra-cluster dispersion.

### 4.5 INTERPOLATION

Samples $\mathbf{x}^{(i)}, \mathbf{x}^{(j)}$ are mapped via the encoder $E$ to representation vectors $v^{(i)}, v^{(j)} = E(\mathbf{x}^{(i)}), E(\mathbf{x}^{(j)})$, and finally to nodes $\mathbf{n}_i$ and $\mathbf{n}_j$ in the SOM tensor via cluster assignment. The optimal interpolative path is discovered via Dijkstra's algorithm. See Figure 2 for a visual representation.

The smoothness of the interpolation is a function of the $\sigma$ parameter in the self-organizing map, as this parameter determines how finely the map partitions the set of encodings into clusters. The self-organizing map can be retrained with a smaller $\sigma$ if smoother interpolations are desired.

$$v^{(i)}, v^{(j)} = E(x^{(i)}), E(x^{(j)})$$

$$n_0, n_5 = BMU(v^{(i)}), BMU(v^{(j)})$$

$$\text{Dijkstra}(n_0, n_5) = (n_0, n_1, n_2, n_3, n_4, n_5)$$

where $w_{e_{ab}}$ weights edge $(n_a, n_b)$

Figure 2: Interpolation between the samples $x^{(i)}, x^{(j)}$ within representation space $\mathcal{X}$ is optimized within the SOM grid by running Dijkstra's shortest path algrothm between the best matching unit weight vectors $w_{\text{SOM}}^{(i)}, w_{\text{SOM}}^{(j)}$ corresponding to the samples $v^{(i)}, v^{(j)}$, respectively.

Because the nodes in the SOM index weight vectors that do not necessarily correspond to actual embeddings, we must perform a nearest neighbors query to recover a representative embedding. For the data considered here, we can compute brute-force nearest neighbors. If operating at scale, this can be accomplished with a vector database algorithm such as FAISS or SCANN.

## 5 EXPERIMENTS

Training FedSOM on a given dataset is a multi-step process. Hyperparameter search is performed for embedding creation (NN and UMAP) and separately for SOM and FedSOM training via the tree-structured Parzen estimator in Optuna. Full hyperparameter ranges for every variable and configs containing optimal hyperparameters for both encoders, SOM, and FedSOM models are available in the code. We experiment on a variety of tabular datasets of widely varying type, with a focus on security datasets, including Ember and Sorel20M. Every FedSOM model consists of at least two constituent SOMs. We compare SOM and FedSOM unsupervised clustering performance on both UMAP and encoder-generated representations by computing normalized mutual information (NMI) and adjusted rand score (ARS). Each MNIST-like image contains a caption giving label as well as graph coordinates. Malicious binary file moduli spaces for three malware datasets show how various malware families relate to each other from the point of view of the leveraged parser.

Table 1: Comparison of Methods on MNIST and Fashion-MNIST Datasets

| | NMI | |
| --- | --- | --- |
| **Method** | **MNIST** | **Fashion-MNIST** |
| k-means | $0.541 \pm 0.001$ | $0.545 \pm 0.000$ |
| minisom | $0.342 \pm 0.012$ | $0.475 \pm 0.002$ |
| GB-SOM | $0.519 \pm 0.005$ | $0.514 \pm 0.004$ |
| VQ-VAE | $0.409 \pm 0.065$ | $0.517 \pm 0.002$ |
| no_grads | $0.001 \pm 0.000$ | $0.018 \pm 0.016$ |
| gradcopy | $0.436 \pm 0.004$ | $0.444 \pm 0.005$ |
| SOM-VAE | $0.594 \pm 0.004$ | $0.590 \pm 0.003$ |
| FedSOM | **0.883** | **0.613** |

## 6 CONCLUSION

A self-organizing map trained in a federated way can effectively cluster both simple image data as well as tabular cybersecurity data. FedSOM trained on parsed malicious binaries can also serve as a moduli space of malware families in which family similarity is presented in terms of proximity within the organizing tensor. This moduli space can be leveraged for interpolation between individual samples and between entire clusters via the underlying weighted graph structure, which

Table 2: Performance of SOM and FedSOM on MNIST-like and Security Datasets

| Dataset | UMAP | | Encoder | | Model |
|---|---|---|---|---|---|
| | NMI | ARS | NMI | ARS | |
| **MNIST-like Datasets** | | | | | |
| mnist | **0.889** | **0.872** | 0.580 | 0.349 | SOM |
| | 0.883 | 0.824 | 0.574 | 0.398 | FedSOM |
| fashionmnist | **0.684** | **0.515** | 0.270 | 0.020 | SOM |
| | 0.613 | 0.483 | 0.194 | 0.066 | FedSOM |
| chars74k | **0.677** | **0.333** | 0.485 | 0.072 | SOM |
| | 0.631 | 0.303 | 0.332 | 0.092 | FedSOM |
| cifar10 | 0.095 | 0.047 | **0.130** | 0.004 | SOM |
| | 0.113 | **0.058** | 0.073 | 0.040 | FedSOM |
| emnist | 0.736 | 0.537 | 0.447 | 0.062 | SOM |
| | **0.746** | **0.564** | 0.390 | 0.236 | FedSOM |
| kuz | 0.602 | 0.380 | 0.346 | 0.043 | SOM |
| | **0.610** | **0.402** | 0.322 | 0.141 | FedSOM |
| notmnist | 0.568 | 0.322 | 0.447 | 0.163 | SOM |
| | **0.596** | **0.475** | 0.361 | 0.143 | FedSOM |
| quickdraw | 0.602 | 0.380 | 0.346 | 0.043 | SOM |
| | **0.610** | **0.402** | 0.322 | 0.141 | FedSOM |
| slmnist | 0.568 | 0.322 | 0.447 | 0.163 | SOM |
| | **0.596** | **0.475** | 0.361 | 0.143 | FedSOM |
| **Security Datasets** | | | | | |
| ember | 0.356 | 0.112 | **0.662** | 0.548 | SOM |
| | 0.327 | 0.120 | 0.551 | **0.570** | FedSOM |
| ccc | **0.333** | 0.059 | 0.295 | 0.071 | SOM |
| | 0.243 | 0.014 | 0.236 | **0.076** | FedSOM |
| pdfmalware | **0.291** | **0.239** | 0.132 | 0.027 | SOM |
| | 0.209 | 0.120 | 0.093 | 0.076 | FedSOM |
| sorel | 0.261 | 0.046 | **0.373** | 0.029 | SOM |
| | 0.076 | 0.021 | 0.217 | **0.054** | FedSOM |
| syscalls | **0.321** | 0.042 | 0.286 | 0.020 | SOM |
| | 0.166 | 0.059 | 0.181 | **0.082** | FedSOM |
| syscallsbinders | **0.336** | 0.035 | 0.312 | 0.053 | SOM |
| | 0.233 | 0.069 | 0.190 | **0.073** | FedSOM |

Table 3: UMAP shows superior performance generating representations amenable to both the single SOM and FedSOM models on MNIST-like datasets. The encoder produces representations more amenable to SOM clustering for large security datasets. UMAP is still superior for smaller security datasets. Best NMI and ARS scores for each dataset are bold.

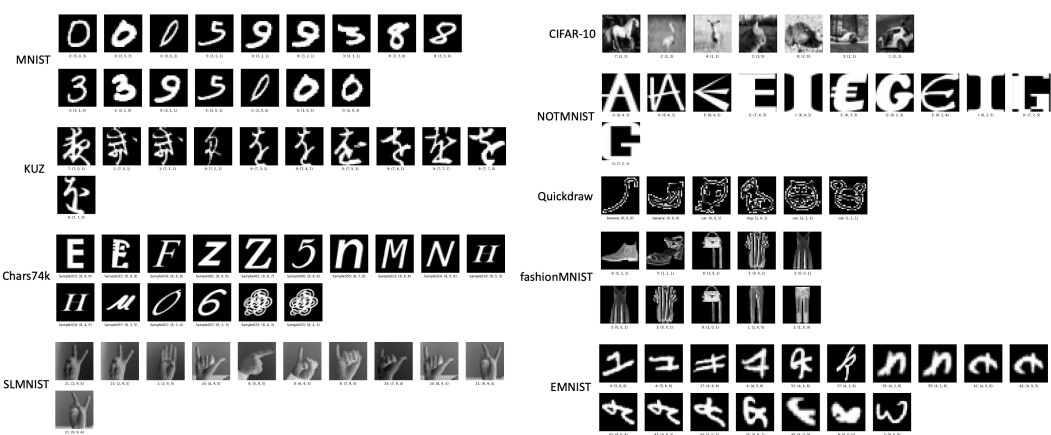

Figure 3: Interpolations occur through clusters in terms of actual datapoints rather than points generated from a latent space via a decoder.

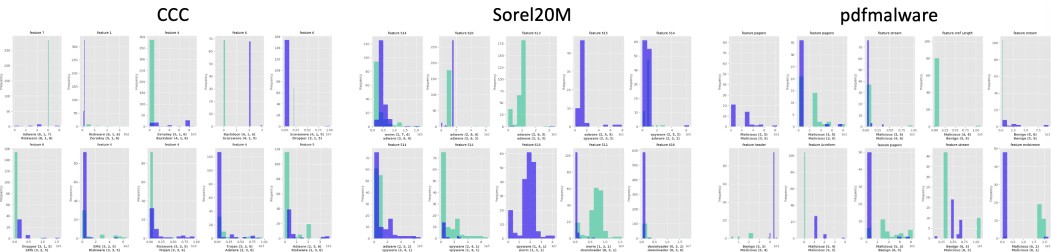

Figure 4: Interpolations through the malware datasets CCCS-CIC-AndMal2020 (CCC), Sorel20M (Ember parser), and CIC-Evasive-PDFMal2022 (PDFMalware). Sequences display the largest successive feature distribution changes along the interpolative path between clusters of various types. This provides analysts the ability to understand the evolution of various feature distributions as a given file class is interpolated into another.

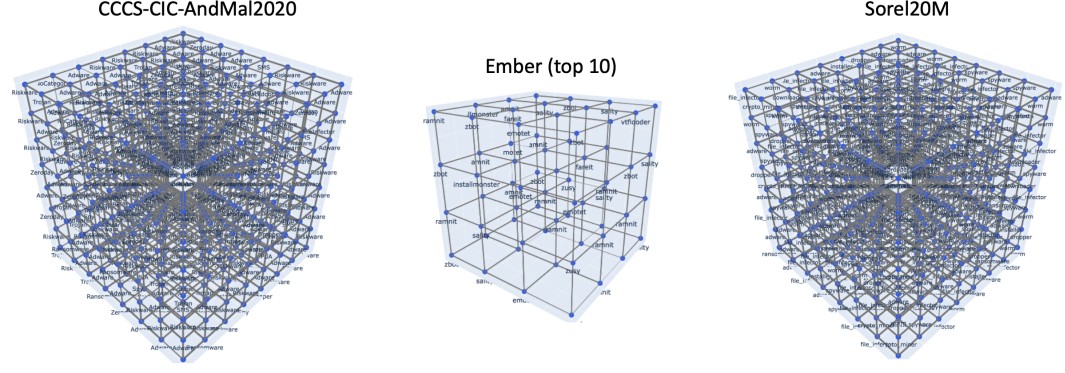

Figure 5: We present discrete moduli spaces for CCCS-CIC-AndMal2020 (CCC), Sorel20M (Ember parser), and Ember. Thicker edges indicate lower cost for the purpose of Dijkstra's algorithm. The moduli space allows analysts to understand the relationships between various file and malware classes, as well as perform nearest neighbors search conditioned on nodes neighboring the node of inquiry.

allows for a deeper understanding of feature distribution changes as the interpolative path is traversed between classes. The samples obtained via interpolation in this way are also faithful to the data generating process, as all such samples belong to the given dataset.

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

## A  APPENDIX / SUPPLEMENTAL MATERIAL

### A.1  NEURAL NETWORK ENCODER ARCHITECTURE

Let $W_i^c$, $W_j^n$, and $W_k^j$ represent the fully connected layers for the categorical, numerical, and joined representations, respectively. The encoder is given by xs

$$\text{representation} = \text{LReLU}(W_1^j([z_{\text{num}}, z_{\text{cat}}])), \tag{3}$$

where we pass $x_{\text{num}}$ through two separate layers $f^{(1)} = W_1^n \mathcal{B}$, $f^{(2)} = \sigma(W_2^n \mathcal{B})$ and then a final layer $f^{(3)} = LW_3^{\text{num}}$, where $\mathcal{B}$ is batch norm and $L$ is layer norm. The portion of the representation constructed solely from numerical columns is then given by $z_{\text{num}} = f^{(3)}(f^{(1)}(x_{\text{num}}) \odot f^{(2)}(x_{\text{num}}))$.

The categorical portion of the representation is given by $z_{\text{cat}} = g^{(3)}(g^{(1)}(x_{\text{cat}}) \odot g^{(2)}(x_{\text{cat}}))$, where $g^{(1)} = W_1^c \mathcal{E}$, $g^{(2)} = \sigma(W_2^c \mathcal{E})$, and $g^{(3)} = LW_3^c$.

We apply the dataset-dependent loss function to the embeddings produced by the projection head

$$\text{embedding} = \text{LReLU}(W_2^j(\text{representation})) \tag{4}$$

## A.2 THE SELF-ORGANIZING MAP

---

**Algorithm 2:** Self-Organizing Map

---

**Result:** $\tau, W = \text{som}(D)$, where $\tau$ is the organizing tensor $\tau$ with integer coordinates and $W$ the set of weight vectors, one for each node of $\tau$

**Data:**

- $D$: the dataset
- $s$: the current iteration
- $\lambda$: number of epochs
- $\eta$: number of iterations per epoch
- $v$: target input data vector
- $i$: index of a node in the map
- $w_i$: weight vector attached to node $i$
- $\beta$: index of the best matching unit (BMU)
- $\theta(\beta, i, s)$: the kernel function
- $\alpha(s)$: learning rate

**for** $s$ *in* $[1, \dots, \lambda]$ **do**
    Sample batch $x \sim D$;
    **for** $t$ *in* $[1, \dots, \eta]$ **do**
        **for** $v$ *in* $x$ **do**
            Find best matching unit $i$ for $v$;
            $w_i^{t+1} := w_i^t + \theta(\beta, i)\alpha(v - w_i^t)$;
        **end**
    **end**
**end**

---

## A.3 DIJKSTRA'S ALGORITHM

---

**Algorithm 3:** Dijkstra's Algorithm

---

**Require:** $G$ (weighted graph), $u$ (source node), $v$ (destination node)
 1: Initialize `distances` with infinity for all nodes except $u$
 2: `distances[u]` $\leftarrow 0$
 3: Initialize `priority_queue` with $(0, u)$ {Priority queue with (distance, node)}
 4: **while** `priority_queue` is not empty **do**
 5:    $(\text{current\_distance}, \text{current\_node}) \leftarrow$ extract min from `priority_queue`
 6:    **if** `current_node` equals $v$ **then**
 7:       **return** `distances[`$v$`]` {Shortest path found}
 8:    **end if**
 9:    **if** `current_distance` $>$ `distances[current_node]` **then**
10:       **continue**
11:    **end if**
12:    **for** each `neighbor` of `current_node` **do**
13:       $\text{distance} \leftarrow \text{current\_distance} + \text{weight}(\text{current\_node}, \text{neighbor})$
14:       **if** `distance` $<$ `distances[neighbor]` **then**
15:          `distances[neighbor]` $\leftarrow$ `distance`
16:          insert $(\text{distance}, \text{neighbor})$ into `priority_queue`
17:       **end if**
18:    **end for**
19: **end while**
20: **return** **None** {No path found}

---

Early experiments were run with $\omega(e_{uv}) = \kappa/(1 + \kappa)$ based on an assumption that interpolation between close and disperse clusters would result in smoother transitions than interpolation between

concentrated and separated clusters. However, pairwise disperse and close clusters were often both ontologically and structurally heterogenous, making interpolation less meaningful.

## A.4 Losses

### A.4.1 C3

The C3 loss (fill in reference) is given by

$$\mathcal{L}_{C3} = \frac{1}{2N} \sum_{i=1}^{N} (\tilde{\ell}_i^a + \tilde{\ell}_i^b)$$

where $\tilde{\ell}_i^a$ (sim $\tilde{\ell}_i^b$) are given by

$$\tilde{\ell}_i^a = \log \frac{\sum_{k \in \{a,b\}} \sum_{j=1}^{N} \mathbf{1}\{z_i^{a\top} z_j^k \geq \zeta\} \exp(z_i^{a\top} z_j^k)\}}{\sum_{k \in \{a,b\}} \sum_{j=1}^{N} w_{ij}^k \exp(z_i^{a\top} z_j^k)},$$

and

$$w_{ij}^k = \frac{\exp(\Gamma(1 - |z_i^{a\top} z_j^k|))}{\sum_{k \in \{a,b\}} \sum_{j=1}^{N} \exp(\Gamma(1 - |z_i^{a\top} z_j^k|))}$$

### A.4.2 Contrastive Loss

Of the many varieties of contrastive loss, we leverage the form described in (ref Hinton or SAINT), i.e.,

$$\mathcal{L}_{\text{cont}} = -\sum_{i=1}^{m} \log \frac{\exp(z_i \cdot z_i'/\tau)}{\sum_{k=1}^{m} \exp(z_i \cdot z_k'/\tau)}, \tag{5}$$

where $z_i, z_i'$ are the representations of inputs $x_i$ and its perturbation $x_i'$, respectively.

