# OpenReview forum: "Deep Clustering and Interpolation via the Federated Self-Organizing Map"
_ICLR.cc/2025/Conference — ICLR 2025 Conference Withdrawn Submission_

### Official Review · Reviewer_jGwM · 2024-11-01

**Soundness:** 1
**Presentation:** 2
**Contribution:** 1
**Rating:** 3
**Confidence:** 5

**Summary:**

This article introduces a clustering and interpolation module based on the self-organizing map (SOM). FedSOM can be added to any encoder and can be trained in a federated manner.

FedSOM creates a discrete space of representations, allowing for interpolation at the cluster. Interpolation is performed by considering the underlying tensor of the SOM as an undirected weighted graph, where the weights are calculated based on the dispersion of the two clusters corresponding to the nodes defining the given edge.

**Strengths:**

-Creation of a Discrete Space: FedSOM creates a discrete modular space of representations, enabling interpolation at the cluster or sample level

-Graph-Based Interpolation: Interpolation is achieved by considering the underlying SOM as an undirected weighted graph.

FedSOM can be trained in a federated manner, meaning different SOM models can be learned on different parts of the data, then combined to form a global model.

**Weaknesses:**

- Difficult to pinpoint the contribution: Is it a contribution in deep clustering, in federated learning, ....

- Sensitivity to encoder perturbations: small changes in the encoder can modify the inferred metric on the representation space, potentially altering the interpolation.

- Section 4.1.2 requires more details on the cost function. In my opinion, it's not enough to simply refer to an original reference.

 The SOM cost function term should also appear in the cost function (with the neighborhood function).

- Section 3.1 , the other key SOM hyperparameter  is the size  of the SOM map

- FedSOM seems to me as an ensemble clustering approach. Ideally, the centroid should be adjusted by the cardinality of observations captured in each local SOM before using them in the meta-SOM.

- I notice a complete disregard for deep SOM approaches.

SOM-CPC: Unsupervised Contrastive Learning with Self-Organizing Maps for Structured Representations of High-Rate Time Series
https://paperswithcode.com/search?q=author%3AArthur+A.+Nijdam

Forest, F., Lebbah, M., Azzag, H. et al. Deep embedded self-organizing maps for joint representation learning and topology-preserving clustering. Neural Comput & Applic 33, 17439–17469 (2021). https://doi.org/10.1007/s00521-021-06331-w

J Xie, R Girshick, and A Farhadi (2016) Unsupervised deep
embedding for clustering analysis. In: International conference on machine learning (ICML), vol. 48, arXiv:1511.06335

M Pesteie, P Abolmaesumi, and R Rohling (2018) Deep neural
maps. In: ICML workshop. arXiv:1810.07291

Ferles C, Papanikolaou Y, Naidoo KJ (2018) Denoising autoen-
coder self-organizing map (DASOM). Neural Netw 105:112–131.
https://doi.org/10.1016/j.neunet.2018.04.016

L Manduchi, M Huser, G Ratsch, and V Fortuin (2020) DPSOM:
deep probabilistic clustering with self-organizing maps. arXiv:1910.01590

**Questions:**

- Explain the role of the tabular disturbance module.

- Why not use the batch version of SOM, which is better suited to the federative approach?

-Even though SOM is a clustering algorithm, it is typically followed by another clustering method, such as hierarchical clustering or K-means, as SOM produces several micro-clusters. For example, we compare SOM+kmeans with the results of K-means alone.  The som generally provides several micro clusters by constructing a topology.

---

### Official Review · Reviewer_4QHo · 2024-11-02

**Soundness:** 3
**Presentation:** 3
**Contribution:** 3
**Rating:** 3
**Confidence:** 3

**Summary:**

This paper introduces FedSOM, a clustering and interpolation module based on the Self-organizing Map. Specifically, FedSOM can be appended to any encoder and trained in a federated way. Finally, it is validated on MNIST-like and parsed-binary malware datasets.

**Strengths:**

- Self-organizing maps are graph structures, which is new.
- This paper introduces a self-organizing map structure that can be trained in a federated way.

**Weaknesses:**

-	This paper is poorly written, for example, there is no period at the end of the contribution point in the introduction. Besides, the figures and tables related to the experiment are concentrated on the last two pages.
-	The experimental dataset is too simple and needs to be tested on a larger dataset, such as ImageNet.
-	Comparison of the computational cost of the proposed algorithm and existing algorithms is needed.

**Questions:**

I am a little confused as to why need to train in a federated way. What is the realistic scenario?

---

### Official Review · Reviewer_9LJU · 2024-11-03

**Soundness:** 2
**Presentation:** 1
**Contribution:** 1
**Rating:** 3
**Confidence:** 5

**Summary:**

This paper introduces FedSOM, a federated clustering and interpolation framework based on Self-Organizing Maps (SOMs). FedSOM can be integrated with different encoders to generate a moduli space of data representations that supports hierarchical clustering and interpolation. The framework aims to enable cluster-level interpolation by modeling clusters as a graph with nodes weighted by inter-cluster distances, where Dijkstra’s algorithm is used to find optimal paths between clusters.

**Strengths:**

1. This paper extends traditional SOM clustering to a federated setting, allowing SOMs to be trained on distributed data without sharing sensitive information.
2. The technical details are properly illustrated and easy to follow.
3. The authors validate FedSOM on cybersecurity data and MNIST-like image datasets, highlighting its clustering performance and interpolation capabilities.

**Weaknesses:**

1. The proposed FedSOM essentially extends traditional Self-Organizing Maps by applying them in a federated setting. However, this "federated" component lacks substantial methodological novelty. The federated aspect here involves independently training multiple SOMs on distributed subsets and then combining their weights in a meta-SOM. This approach does not appear to incorporate any sophisticated techniques to handle common federated learning challenges, such as handling non-IID data or communication efficiency.
2. The experiments are limited primarily to cybersecurity data and MNIST-like datasets, which limit the generalizability of FedSOM.  Since the authors claim the framework can handle diverse data types, additional experiments on more complex or heterogeneous datasets (e.g., non-image, tabular data, graph) are essential to support its model-agnostic claims.
3. The authors stated that the SOM tensor dimensionality grows exponentially, which could introduce significant computational costs, especially as data size increases. However, this paper does not include any computational complexity analysis, which makes it difficult to assess the scalability of FedSOM.
4. The reproducibility is very poor. The reviewer could not find the detailed experimental setup or reproducible source-code for the proposed method.
5. The paper primarily compares FedSOM with older baselines, lacking the latest state-of-the-art methods. The comparisons with more advanced methods are essential to substantiate FedSOM’s effectiveness.

**Questions:**

See weakness.

---

### Official Review · Reviewer_eqmf · 2024-11-04

**Soundness:** 2
**Presentation:** 1
**Contribution:** 1
**Rating:** 3
**Confidence:** 4

**Summary:**

Proposes an approach for deep clustering based on the use of self organizing maps.   A distinctive aspect of the approach is that a federated method can be used for training.  A particular focus is placed on application of the approach in the context of cybersecurity data.

**Strengths:**

Deep clustering is an important direction, of interest to many in the ICLR community.  Practitioners of SOMs may appreciate new methods for their use in the deep clustering area.

**Weaknesses:**

-The running time of the approach is not explored experimentally or compared with baseline approaches.  How does federated training time vary according to the number of SOMs?

-Discussion of experimental results is lacking (nothing for tables 1,2,3 and figures 3,4,5), they are not analysed in the text.  Is evaluation using NMI/ARI with ground truth cluster labels – are these the classes?  There is a lack of concrete detail about hyper parameters used in the experiments.   Details about dataset characteristics have not been provided and no rationale is given as to why they were chosen.  No ablation study is provided.

-cybersecurity data is mentioned at different points in the paper, but there isn’t a discussion of what particular challenges this type of data brings for deep clustering and how federated SOM overcomes these challenges for this type of data.  Is the method less suitable for non cyber security data?     If cyber security is a focus, how actionable are the discovered clusters in a cyber security context.

**Questions:**

Please consider/address comments under weaknesses.

---

### Official Review · Reviewer_Yirs · 2024-11-05

**Soundness:** 2
**Presentation:** 1
**Contribution:** 2
**Rating:** 3
**Confidence:** 2

**Summary:**

Authors propose clustering and interpolation method based on Self Organizing Maps (SOM) that can be trained in the federated learning setting. Proposed method FedSOM consists of multiple SOMs trained on the subset of data in the federated learning manner. Meta-SOM is learned on weight representations of each local SOM which is further used for clustering and interpolation. Method builds a graph using SOM nodes, and edges represent connections between neighboring nodes in the grid. Furthermore, proposed method can interpolate between two clusters or samples, which can be determined using Dijkstra's algorithm.

**Strengths:**

Authors proposed federated training approach for SOMs

**Weaknesses:**

The novelty of the work is very small. The work is rather a heuristic way to extend a SOM algorithm.

**Questions:**

N/A

---

### Comment · Area_Chair_hX2D · 2024-11-13
**authors - reviewers discussion open until November 26 at 11:59pm AoE**

Dear authors & reviewers,

The reviews for the paper should be now visible to both authors and reviewers. The discussion is open until November 26 at 11:59pm AoE.

Your AC

---

### Note · Authors · 2024-11-18

I have read and agree with the venue's withdrawal policy on behalf of myself and my co-authors.